# Adaptation to Disaster Risk—An Overview

**DOI:** 10.3390/ijerph182111187

**Published:** 2021-10-25

**Authors:** Huicong Jia, Fang Chen, Enyu Du

**Affiliations:** 1Key Laboratory of Digital Earth Science, Aerospace Information Research Institute, Chinese Academy of Sciences, Beijing 100094, China; jiahc@radi.ac.cn; 2College of Resources and Environment, University of Chinese Academy of Sciences, Beijing 100049, China; duenyu@cug.edu.cn; 3Hainan Key Laboratory of Earth Observation, Institute of Remote Sensing and Digital Earth, Chinese Academy of Sciences, Sanya 572029, China; 4State Key Laboratory of Remote Sensing Science, Aerospace Information Research Institute, Chinese Academy of Sciences, Beijing 100094, China

**Keywords:** disaster risk, climate change, adaptation, method, digital disaster reduction

## Abstract

The role of natural disaster adaptation is increasingly being considered in academic research. The Paris Agreement and Sustainable Development Goal 13 require measuring the progress made on this adaptation. This review summarizes the development stages of adaptation, the multiple attributes and analysis of adaptation definitions, the models and methods for adaptation analysis, and the research progress of natural disaster adaptation. Adaptation research methods are generally classified into two types: case analysis and mathematical models. The current adaptive research in the field of natural disasters focuses primarily on the response of the social economy, especially the adaptive decision making and risk perception at farm-level scales (farmer households). The evaluation cases of adaptation in the field of disasters exist mostly as a part of vulnerability evaluation. Adaptation and adaptive capacity should focus on four core issues: adaptation to what; who or what adapts; how does adaptation occur; what is adaptation; and how good is the adaptation. The main purpose of the “spatial scale–exposure–vulnerability” three-dimensional scales of adaptation assessment is to explore the differences in index system under different scenarios, the spatial pattern of adaptations, and the geographical explanation of its formation mechanism. The results of this study can help and guide future research on integrating climate change and disaster adaptations especially in regional sustainable development and risk reduction strategies.

## 1. Introduction

Since the proposal of climate change in the 1970s and its impact on human society, the international scientific community and governments began discussing how human society should respond to global changes and adopt corresponding countermeasures. The specific research direction proposed in the 1970s was prevention, with mitigation in the 1980s, and adaptation to date. Prevention, mitigation, and adaptation are all human response behaviors [1,2,3,4,5]. The term “adaptation” is currently used in the climate field. It originated from natural science in the field of population biology and evolutionary ecology [6]. It originally referred to the general characteristics that ensure the survival and reproduction of organic individuals in living environments. These characteristics result in sustainable survival and the development of species or ecosystems, while evolving to changes in an organism or species makes it more adapted to survive [7]. The IHDP (International Human Dimensions Program) launched the “Inter-vulnerability framework to assess interacting impacts of global processes” in January 2005, which proposed dynamic changes in time and space to form the vulnerability and process of climate change and that of globalization [2,8]. A vulnerability evaluation concept combined with a global change process proposes to transform the general index evaluation methods in most studies into the vulnerability evaluation of the adapter. The vulnerability of the adapter is not only a function of the exposure level, sensitivity, and adaptability but also includes the adaptors’ cognitive process for changes, risks, trade-offs, and the selection of adaptation methods [9].

Through the 2015 Paris Agreement on Climate Change, 197 countries have committed to 47 ambitious efforts to combat climate change, adapt to its effects, and provide enhanced support to developing countries [10]. Enhancing understanding and management of risks, as well as the impacts that impinge upon individuals, households, communities, cities, countries, economies, or ecologies through time, is at the heart of the aspirations and goals of the Sendai Framework, which was adopted by Member States at the United Nations General Assembly in June 2015 [11]. In 2015, UN member countries also adopted the 2030 Agenda for Sustainable Development—a comprehensive global plan of action for “people, planet and prosperity” comprised of 17 Sustainable Development Goals (SDGs) and 169 targets to be achieved by 2030, including Goal 13 on climate action [12]. The goal of “Climate Action: Take urgent action to combat climate change and its impacts” (SDG 13) is to reduce the impact of climate change on people and improve the ability to respond to climate change. An important way to reinforce SDG 13.1 (Strengthen resilience and adaptive capacity to climate-related hazards and natural disasters in all countries) is to explore and analyze the connotation and assessment methods of adaptation. In this review, we fill this gap by providing an overview of adaptation and adaptive capacity evaluation methods and their applications in the fields of climate change and natural disaster risk reduction.

In a changing climate, disasters are inevitable due to the uncertainty and abnormality of hazards and the expanded exposure. Over the past three decades, evidence has mounted that the global climate is changing and that anthropogenic greenhouse gas emissions are largely to blame [12]. According to the Intergovernmental Panel on Climate Change (IPCC), climate change includes increasing temperatures, changing rainfall patterns, rising sea levels, saltwater intrusion, and a higher probability of extreme weather events that could lead to natural disasters [10,12]. The length, frequency, and/or intensity of heat waves will increase with higher temperatures when extremes occur. Heavy rainfalls associated with tropical cyclones are likely to increase with continued warming. The intensity and frequency of extreme precipitation events are very likely to increase over many areas, and the return period of extreme rainfall events is projected to decline, resulting in more numerous floods and landslides. Mid-continental areas will generally become dryer, which is likely to increase the risk of summer droughts and wild fires.

This paper is organized as follows: after the Introduction section, the definition of adaptation and development stages of adaptation are discussed in Section 2 and Section 3. Section 4 and Section 5 summarizes analysis methods of adaptations and regional applications for different fields of the natural sciences. Finally, Section 6 provides a summary and discussions.

## 2. Definition of Adaptation

### 2.1. Definition of Adaptation and Multiple Properties

Understanding the concept of adaptability has undergone a change process to include processing power and the response to adjustments. Although these definitions have respective focuses, they all emphasize the need to adjust the system and reduce its vulnerability to improve and strengthen its ability to adapt to climate change. The content of adaptation involves the process of natural and sudden disaster impact assessments, which includes countermeasures against climate change to enhance the process of designing and improving measures for sustainable regional development [13].

The definition of “adaptation” has many attributes, including the two most important points (Table 1). First is the spatial scale of adaptation, which depends on who is responsible. Second is the nature of adaptive behavior, whether it is spontaneous or conscious or it is planned or prescriptive. The former is usually short-term and tactical adaptation, which is directly related to specific climate change. The latter is more strategic, long-term, and proactive and is usually formulated by government departments and used as part of policy adaptation measures [14]. The adaptation to climate change in the literature is sometimes divergent at the temporal and spatial scales. Short-term adaptation is more of a reaction, and higher-scale adaptation is considered an expected adaptation through policies, projects, and recent plans and actions [15].

Smit and Skinner further defined several key characteristics of adaptability, including the purpose, time and duration, scale and responsibility, and form of adaptation [20]. At the same time, several adaptive approaches have been proposed, such as technological development, government projects and insurance, production reform, and financial management, which provide a useful framework for the development and selection of adaptive strategies toward human vulnerability. Fankhauser et al. proposed three elements of adaptation: necessity, motivation, and capability [36]. Bryant et al. gave four main components of adaptability: pressure characteristics, system characteristics (including cultural, economic, political, institutional, and biophysical environment), multi-scale, and adaptive response [37]. The different spatial scales (from local adjustments to regional and national resource reorganization strategies and policies) and temporal scale changes (from short-term changes to longer time scales) cause adaptations to vary widely [34]. Climate change is only one aspect of response and adaptation, which is closely related to other human and economic factors.

### 2.2. Analysis of Adaptation-Related Terms

As an important attribute of disaster systems, adaptation is closely related to the concepts of other disasters; however, they exhibit differences when used.

#### 2.2.1. Adaptation, Adaptability, and Capacity of Response

Adaptability in ecology is the ability to adapt to certain environmental changes, while adaptation is the characteristic of structure, function, and organizational behaviors [38]. Adaptability is the external manifestation of adaptive ability and shows a way to reduce vulnerability [18]. A system’s capability to better handle exposure and sensitivity reflects the capability to adapt [39] (Figure 1). Many adaptive forms and levels can be divided based on timing (anticipated, current), intention (automatic, planned), spatial scales (local, wide-area), and form (technical, behavioral, financial, economic, institutional, and information) [40]. The adaptation of the original system can be distinguished from the degree of adjustment [41].

Local adaptive capability is a comprehensive reflection of several conditions [42,43], which is reflected by factors such as management capabilities, economic and financial conditions, technology and information resources, infrastructure, and institutional environments [16,44,45,46,47,48]. In general, improving environmental conditions allows a species, population, or individual to better adapt to the environment. Due to the human field and social ecosystems, the standards of adaptation far exceed the ability to survive and reproduce, which includes the results of social and economic activities and the quality of life [39]. Smithers and Smit noted that the adaptability of the human system includes the capability of the social ecosystem to respond to environmental changes and to promote improved conditions related to the environment [49].

Kasperson et al. distinguished between adjustment and adaptation [50]. They believed that adjustments are a system’s response to interference or pressure without fundamentally changing the system itself. This is a short-term and relatively small system adjustment as adaptation is the system’s response to interference or pressure. The response to stress changes the system itself and can sometimes transform a system state to a new state [51]. The concepts involved in adaptability include the coping ability, management capacity, stability, robustness, and flexibility [28,49,52,53,54]. From the definition of ISDR (International Strategy for Disaster Reduction), the coping ability refers to all available forces and resources in a community or organization that can reduce the risk level or impact of a disaster [55]. Brooks et al. defined adaptive capability as natural, economic, institutional, or human resources that can be used for adaptation and included the availability of information, professional knowledge, social networks, and other resources [56]. Cultural values also play an important role in the construction of human adaptability [57,58].

The IPCC analyzed the relationship between the adaptability and capacity of a response for the social ecosystem and believed that the connotation of adaptability should be broader than that of the capacity of response. However, these all depend on the specific definition of adaptability and capacity of response in studies of coupled social ecosystems [58]. Adger [59], Smit and Wandel [39], and IPCC [19] all defined the system’s coping ability or capacity of response as adaptability. Turner et al. distinguished the capacity of response from adaptability, considered that both are components of the system resilience, and regarded adaptation as a manifestation of the reconstruction of the system after a response [60]. Generally, the capacity of response is an inherent attribute of the system that describes its ability to respond to interference, mitigate potential damage, take advantage of opportunities, adjust to system changes, and respond to system transformation. The capacity of response is also an attribute that the system has priority over interference [58]. Vogel believed that the coping ability is a short-term behavior only for survival, while the adaptive ability is used for long-term or more continuous adjustments [61].

Boundaries between the medium- and long-term scales are blurred. The response refers to the actions taken by society or individuals when faced with the adverse consequences of climate change or natural disasters, which are considered short-term adjustments to extreme events [59]. Coping strategies usually occur naturally and cause varying degrees of vulnerability. Adaptation includes the stress response, such as changing sources of income, immigration, or other lifestyle changes, as well as long-term intervention by government agencies [62].

#### 2.2.2. Adaptability, Vulnerability, and Resilience

Due to diversity and differences of views, the relationship between adaptability and resilience is unclear. According to Smit and Wandel [39], some scholars have equated adaptability with social resilience. Gunderson et al. regarded adaptability as the effectiveness of a system to changes in resilience [63]. Carpenter et al. regarded adaptability as a component of resilience, which reflects the response of system behaviors to disturbances [64]. Adger regarded adaptability as the collective ability of human activists to manage resilience, which includes reducing or eliminating undesirable factors, creating new expectation factors, and promoting the transformation of the current system to the desired state [59]. Folke et al. believed that vulnerability is the opposite or antithesis of resilience [65]. However, the vulnerability of a system with resilience is lower than that of a system without resilience. Resilience is related to the capacity of response in a vulnerable element, making it smaller than the negative range of vulnerability [58].

The most fundamental difference is that resilience is applied to the maintenance of system behaviors, and the opposite of vulnerability is the ability to resist interference and maintain the system structure. Therefore, for the elements of social ecosystems, resilience appears to be a true subset of the adaptive capacity. Adaptability includes not only the resilience of the system but also its ability to cope with impacts and take advantage of opportunities [66]. Adaptation is a measure taken by humans with a constant evaluation of vulnerability. In the formula V_ist_ = f(E_ist_, A_ist_) as modified by Smit and Pilifosova [42], V is the vulnerability, E is the exposure sensitivity, and A is the adaptive capacity. Here, i refers to the system, s refers to the climate stimulus, and t is time. The sensitivity refers to the degree to which the system suffers and responds to climate stimuli. Exposure is a factor of risk that is the total amount of hazard-affected bodies that are exposed to a hazard. Gallopin [58] used a systematic perspective to comprehensively analyze the relationship between the concepts of vulnerability, resilience, and adaptability (Figure 2), which has been adopted by some scholars [2,3,67,68].

#### 2.2.3. Adaptation and Mitigation

Climate change risks can be managed through efforts to mitigate climate change forcers, adaptation of impacted systems, and remedial measures. Mitigation refers to efforts to reduce or prevent the emission of greenhouse gases or to enhance the absorption of gases already emitted, thus limiting the magnitude of future warming [33]. Mitigation avoids difficult to handle situations, while adaptation aims to manage the inevitable consequences [69]. There are uncertainties in adaptive strategies that require long-term perspectives, which may be unpopular for current governments. The current understanding of adaptation is to regard the adaptation period as placing future social public resources in danger. The adaptation strategies must be concurrent and complementary with mitigation efforts because, over the long term, emissions reduction choices will determine the severity of climate change, its impacts, and the degree of adaptation required in the future [70]. Unlike mitigation, adaptation is most practical at the local level.

## 3. Development Stages of Adaptation

### 3.1. Adaptation in Disasters and Other Fields

The concept of “cultural adaptation” was firstly used to describe the “cultural cores” to the physical environment [71,72]. To date, the adaptive example has been widely used in social sciences. Denevan defined cultural adaptation as the response to natural environmental changes and the associated humanities (such as population, economics, and organization) [71]. O’Brien et al. defined adaptation as an organizational or group enhancement environment or cultural grinding ability and believed that adaptation is the behavior selection result produced by cultural practice in changing environments [73]. In adapting to the subject, biological adaptation involves changes in individuals and populations, while social adaptation is the adjustment of individual and collective behaviors, and there are similarities between the two [49]. However, the ability of human systems to exhibit planned and managed adaptation makes it an important factor to contain natural environments and intrinsic stimulation double changes. This allows adaptation to be combined with environmental perception and risk assessment as important factors in adaptation strategies. In addition, as the human system is culturally adapted, human groups can create new and improved methods to process environmental issues into their culture. Thus, the pursuit of human systems is the adjustment of the target, not just the survival of the species, which includes an enhanced quality of life.

In the field of disasters, some scholars, such as Burton et al. [74], emphasized that adaptation should include risk cognition, adjustment, and disaster management. In addition, many scholars have focused mostly on adaptive adjustments and environmental disaster management [75]. Holling also used the concept of adaptation to study the recovery, balance, and adaptation management of natural environmental changes [76]. In the field of power theory and food safety, adaptation is seen as a resource acquisition and a response to people’s behaviors, which forms a highlighted feature in this research area to reveal how individual or family adaptability is formed and how the social, political, and economic processes can be restrictive [77,78,79]. For adaptive research in political ecology, Kasperson believed that adaptation should be launched to research individual and family adaptability and considered how these can be shaped and restricted by social, political, and economic processes [50,80]. Currently, research on adaptation to disaster risk reduction is still in its initial stages, and there is no unified definition.

### 3.2. Adaptation in Climate Change

Adaptive research has continued to emerge with the constant concern of climate change. Early cases from Butzer [81] were based on predictable climate change and its expected impact on the world food supply while considering cultural adaptation (human wisdom from technological innovation and long-term planning). Since then, the adaptation analysis and research of climate change have gradually expanded [18,42,82]. Smit et al. proposed a schematic diagram for adapting to climate change and variations, which consists of several problems [18]. The Paris Agreement aims to strengthen the ability of countries to deal with the impacts of climate change through appropriate financial flows, a new technology framework, and an enhanced capacity-building framework. Saving lives and livelihoods requires urgent action to address the climate emergency and adaptation [83]. Through scientific guidance (understanding and prediction), the public participation (communication and education), scientific adaptation, adaptation management and methods, and decision-making (global convention and implementation of national strategies) are all interrelated.

## 4. Analysis Models and Methods of Adaptation

The research methods of adaptation are generally classified into two types: case analysis and mathematical models. Vulnerability research tends to focus on constructing and analyzing indicators. Research on resilience, especially in ecology, has developed several theories and mathematical models, while research on adaptation has focused on case analysis [84]. Table 2 summarizes the characteristics of all the methods of adaptation.

### 4.1. Evaluation Standard

Although there have been some discussions on adaptation, such as the NFCCC (United Nations Framework Convention on Climate Change) and UNAPF (United Nations-Azerbaijan Partnership Framework), there is still no agreement on its goals; thus, the success of the adaptation behavior cannot be judged. Mercer [85] believed that the goals of adaptation should include the following aspects: maintaining risks associated with climate change at the current level, reducing risks to a lower level when existing risks are considered unacceptable, reduce the exposure of vulnerable populations, and others. A successful adaptation should consider the following points: cost–benefit, efficiency, distribution of costs and benefits, and legality of adaptation. Other aspects include sustainability, global and intergenerational fairness, and adaptation in harmony with cultural norms and socially recognized values [5,86].

The following situations may appear in some cases of adaptation to climate change. Success or failure is also affected by self-adaptation or adaptive behaviors. Individuals experiencing climate change can often distinguish whether they are well protected and better adapted, indicating there may also be different values that support various adaptation goals [87]. Temporary adaptation will only bring future challenges. Therefore, adaptation should be coordinated with the natural environment, including the consciousness formation of modern civilized society. Thus, it is necessary to improve the adaptability through capacity building at the local level [82].

The systematic analysis of potential adaptation options needs to consider their possible feasibility, cost, profit, effectiveness, execution speed, and the acceptability of relevant fund custodians [34,88]. The analysis primarily includes the following six aspects:

(1) The adaptive capacity. This can be measured by reducing the impact or exposure or by reducing disasters, preventing danger, and improving safety. However, in practice, the complex causal chain induces several problems when estimating the adaptive capacity [89].

(2) Effectiveness of adaptation. This is the extent to which the behavior reaches the goal. However, many issues need to be considered in the evaluation process. First, selecting a particular adaptation may be uncertain under the given circumstances. Second, the utility of an adaptation option introduced by the organizer may depend on other behaviors. Relying on the effectiveness of adaptation measures depends on individual behaviors and may be very difficult to evaluate. Third, the effectiveness of adaptation behaviors may depend on unknown future conditions. Fourth, an adaptation measure is effective as it reduces the impact of climate change or increases one location or period. Any adaptive behavior can potentially create unintended consequences on other natural and social systems [90].

(3) Adaptation efficiency. Adaptation to climate change needs to bear the cost but should generate significant benefits. The scale costs in the individual organization are the implementation measures, including transaction costs and inaccurately estimated costs, as well as benefits that reduce impacts or enhance opportunities. However, the analysis of efficiency adaptation at any scale is more economical than a simple and quantitative cost–benefit comparison [72].

(4) Cost of adaptation. Although there are many possible adaptation measures, it is necessary to understand the conditions that limit adaptability and the costs of improving adaptability. Fankhauser [52] believed that the impact cost is the sum of the costs of adaptation and residual losses. Any comprehensive evaluation of adaptation costs (including profits) not only considers economic indicators but also social welfare and equity. Adaptation costs can be divided into the direct cost of adaptation, cost of adapting to the state of adaptation, and cost of inadaptability [91].

(5) Fairness and legality of adaptation. Fair adaptation can be evaluated from the perspective of the main body whose income decides to adopt adaptation measures. There are many principles of income fairness, including the principle of deservedness and for fairness or need. Each principle has its own power, where the distribution of power in the income system could impact the legitimacy of decisions [22].

(6) A systematic view of adaptation. Adaptation assessment has become more inclusive over time, linking future climate change with current climate risks and other policy concerns [79]. Adaptation includes well-established practices from disaster risk management (e.g., early-warning systems), resource management (e.g., water rights allocation), spatial planning (e.g., flood zone protection), urban planning (e.g., building codes), public health (e.g., disease surveillance), and agricultural outreach (e.g., seasonal forecasts) [92]. The adaptation of farmers may extend beyond climate change and be multifaceted. For example, unstable land ownership affects the ability of farmers to develop agriculture sustainably. Therefore, economic and institutional factors impact adaptation. These factors gradually undermine the way farmers adapt and impact the accuracy and relevance of climate information they feel.

### 4.2. Evaluation Model and Method

The United Nations Framework Convention on Climate Change (UNFCCC) defines adaptability as being in two categories: spontaneous and planned [93]. The following describes general methods to evaluate adaptability for applications in many fields.

#### 4.2.1. Scenario-Driven by Climate Change

To date, most research on the evaluation of climate change impacts and adaptation countermeasures has adopted so-called “scenario-driven” research methods. This approach is represented by the IPCC technical guidelines to evaluate climate change impacts and adaptation countermeasures. This is usually considered a standard research method or approach and consists of the following seven steps:

(1) Define the problem (clear research area, research content, select sensitive departments, etc.);

(2) Choose an evaluation method suitable for most problems;

(3) Select the test method and conduct sensitivity analysis;

(4) Select and apply climate change scenarios;

(5) Evaluate the impacts on biological, natural, and socio-economic systems;

(6) Evaluate spontaneous adjustment measures;

(7) Evaluate adaptation strategies.

The fourth step is the critical part of the entire evaluation process as it is driven by future climate change and socio-economic scenarios. Thus, an assessment of the impacts of climate change on humans and ecosystems can be performed. Once the ecosystem and socio-economic system are warned that the impact of climate change will be affected, these systems or departments spontaneously respond or adapt and reduce the losses caused by climate change through anticipated adaptation measures and countermeasures. This assessment approach represents a routine procedure that requires significant time, energy, and resources to select and apply climate change scenarios and impact assessments. In practice, there is often insufficient time and funds to conduct adaptation countermeasure assessment research.

In the climate change literature, most research has focused on the losses and impacts of climate change on specific aspects of human society and ecosystems. The main purpose of applying simulation models is to establish the future state of the ecosystem in relation to climatic conditions. For example, several different types of simulation models can be used to study the growth rate of crops or forests under various climate scenarios. Then, different adaptation strategies can be evaluated using the ecological simulation models. Changing the corresponding parameters of the simulation model reflects the adoption of certain adaptation countermeasures or measures under different climate change conditions. All applications have to take account of uncertainties in the information. Some of those are related to uncertainties in the climate models and future emissions, others are related to downscaling to the local scale, and still others are related to the lack of consistent data to verify the model at that local scale. The climate system is changing, so uncertainty about extremes is rising. For example, this could include using varied model parameters to indicate the adaptability of some new crops and tree species to future climate change and the development of production technology to adapt to future climate change [94,95,96,97].

#### 4.2.2. Adaptation Decision Matrix

The adaptation decision matrix (ADM) is based on Excel or Lotus and is used to analyze the cost–benefit of adaptation measures. Researchers list policy goals in the upper part of the matrix and the adaptation strategies (including not taking any measures) in the lower part. Through expert diagnosis, research, and analysis, a value (from 1 to 5) is assigned to each adaptation policy to indicate the degree of satisfaction it can achieve for the specific goals under various adaptation strategies.

Researchers also have the authority to set different weights for each policy objective in the evaluation process and perform a weighted sum to calculate the cost when the benefit increases by one unit. For example, Mizina et al. [98] used an adaptive decision-making matrix and expert-scoring method (using arbitrary quantitative ratios and not monetary values) to analyze 12 adaptive factors that affect Kazakhstan’s agriculture and screen out four important factors. Batima [99] evaluated the drought adaptation of pastures in Mongolia using a decision matrix to divide various measures into the three levels of high, medium, and low in terms of long-term effectiveness, short-term benefits, costs, and limitations. This method is useful when many of the benefits generated from the policy objectives are difficult to monetize or cannot be unified. However, conducting in-depth research requires detailed analysis results to provide researchers with basic information as the basis to evaluate and score. Otherwise, the scoring process relies too heavily on subjective judgment; however, if it is used as part of the questionnaire for statistical analysis, this error effect will be significantly reduced.

#### 4.2.3. TEAM (Tools for Environmental Assessment and Management)

To evaluate the possible impacts and consequences of various adaptation countermeasures and planning to select suitable and satisfactory countermeasures, the United States Environmental Program has developed a decision support system software called TEAM (Tools for Environmental Assessment and Management) as a decision-making tool [100]. This system is based primarily on multi-criteria and multi-standard decision-making technologies and uses graphical means and man–machine dialogue to simplify and clarify the evaluation process. This evaluation method is suitable to assess the impact and adaptability of climate change in water resources, coastal areas, and agricultural sectors. The TEAM model is used as an analysis tool for the U.S. government to evaluate climate adaptation countermeasures based on the national unit’s international climate change impact and adaptation countermeasure evaluation project [101].

The entire evaluation and research process based on the TEAM model includes five main steps, each of which provides a certain mechanism and function (Table 3). The first step is to determine the geographic location of the study area. The geographic location is used to analyze the conditions of various natural resources or ecosystems (such as water resources, agricultural systems, and coastal areas) and determine the characteristics of the vulnerability of these systems to climate variability or change. The second step is to select possible adaptation strategies and measures. The TEAM model recommends a series of adaptation countermeasures and measures to reduce the vulnerability of selected systems or departments for analysts or decision-makers. Analysts or decision-makers can select an appropriate number of adaptation countermeasures or add special countermeasures. The third step is to evaluate the adaptation strategies. The evaluation criteria must be carefully determined to comprehensively evaluate the economic, social, resource, and environmental effects of some climate change adaptation policies and programs. In the fourth step, analysts or decision-makers give scores to each standard or indicator based on the performance and benefits of each adaptation countermeasure in this standard. The TEAM model allows users to compare various quantitative data to determine scores. The fifth step is to display the evaluation results. The TEAM model provides users with several approaches, and the analysis results can be presented in a way that is more acceptable to both the general public and decision-makers, such as diagrams. The displayed evaluation results reflect the benefits of different adaptation strategies to various standards so that information on the associated advantages and disadvantages can be given.

#### 4.2.4. Multi-Criteria Evaluation Method

When evaluating adaptation strategies for multi-standard and multi-group participation, the multi-standard evaluation method is a better analysis technique and can be used as an effective tool. In this way, various adaptation strategies can be compared and evaluated in an orderly and systematic manner. Multi-standard evaluation tools can determine satisfactory policies when given a series of possible adaptation policies to handle biological, natural, and socio-economic climatic vulnerabilities. Several methods and tools developed and established in the fields of decision science, multi-standard evaluation, and system analysis can also be used to evaluate adaptation measures. These can effectively link climate change impact assessments with regional sustainability and include goal planning (GP) [102], fuzzy pattern recognition (FPR) [103], neural network (NN) technology [104,105,106], and multi-level analysis process technology [107].

#### 4.2.5. Agent-Based Modeling Method

Agent-based modeling is a useful policy tool to simulate the effects of different adaptation options toward reducing vulnerability, as it allows the representation of not only dynamic changes in climate and markets but also the dynamic adaptive process of different groups of communities on the impacts of these changes. Model simulations of adaptation options under various global change scenarios show that production support significantly reduces future vulnerability only if complemented with appropriate market support. Therefore, policies need to provide a complementary bundle of adaptation measures. These adaptation measures include developing knowledge on climate change impacts and adaptation; strengthening observations; promoting an approach appropriate for the different territories and utilizing legislative and regulatory instruments, implementing risk-reduction strategies in the insurance industry; etc. Lack of funds and information are the most important reasons to not apply available technical adaptation measures (Figure 3).

## 5. Regional Adaptation Evaluation

Adaptation has become one of the focuses of scientific debates in the field of climate change and disaster risk. This includes practical aspects and seeks ways to understand the methods and concepts that are conducive to human intervention [19].

### 5.1. Adaptation Research in the Field of Climate Change

During global research on climate change and its possible impact on human society, prevention was proposed in the 1970s, and activities that occurred as global-scale mitigations were discussed in the 1980s through now. Multi-scale adaptation research is generally accepted, and adaptation has become a major topic in climate change research [108]. In the United Nations Framework Convention on Climate Change (UNFCC), it is emphasized that mathematical–statistical models are used to estimate the impact of climate change at larger scales and under conditions of adaptation and non-adaptation. The purpose is to emphasize problems of risk under the existence of future climate change scenarios [109,110].

Physical climate information addresses how the climate system responds to the interplay between human influence, natural drivers, and internal variability. Knowledge of the climate response and the range of possible outcomes, including low-likelihood, high-impact outcomes, informs climate services—the assessment of climate-related risks and adaptation planning. Physical climate information at global, regional, and local scales is developed from multiple lines of evidence, including observational products, climate model outputs, and tailored diagnostics.

Based on current mainstream international research, climate change adaptation is focused primarily on the following four aspects.

(1) Adaptive assumptions can be adopted based on the impact conditions or parameter changes measured using the climate change scenario model to estimate and predict the effects of different adaptive methods [111]. However, this type of research has not performed actual investigations or verifications of adaptability [36].

(2) Another focus is primarily on the adaptation options and strategies of special systems under climate change conditions. In the UNFCCC [93] clause, it is emphasized that each country should commit to applying appropriate climate change adaptability and structure while implementing effective response strategies to evaluate the advantages and utilities of adaptation and confirm the best strategy [112,113,114].

(3) A third focus is primarily on the relative adaptation of countries, regions, and communities and selection of certain standards, indicators, and variables for comparative evaluation and classification. This type of research is measured based on certain causal relationships and decisive factors. Through some indicators, scores, and grading processes, this evaluates the relative adaptation of a country, a region, and a community [115,116,117] and accumulates the adaptive capacity elements for each system to form the overall evaluation score [55].

(4) Related research focuses primarily on active adaptation practice strategies. To date, research on adaptive practice processes is not ubiquitous. There are not many targeted studies that directly considered the adaptive label or framework, especially at the regional level, while the community level is weaker [118]. However, adaptation at these levels is often the most practical and most complex. Differences in climate conditions, geographic locations, management systems, real estate, public facilities, resource availability, the implantation of local traditional cognition in the decision-making process, etc., all impact adaptation [119,120]. Nevertheless, many scholars in the fields of resource management, community development, regional planning, food safety, livelihood safety, and sustainable development are involved in studying the adaptive time and process.

Adaptation has temporal and spatial scales that range from household adaptation to government policy formulation. However, due to the impact of the spatial resolution, climate change prediction models are not suitable for farmers’ planting management or other related activities and need to be downscaled [121]. Some scholars have also gradually realized the importance of the farm level in decisions to adapt to the process of adaptation, especially to understand climate extremes, and research has begun considering the role of humans when adapting to the impacts of climate change by investigating farmers’ perceptions and risk management choices. Li and Chen [122] discussed the concept of vulnerability, sensitivity, adaptability, and their associated assessment methods. Vulnerability is affected not only by climate sensitivity but also by the structure, functioning, and succession of the system as well as its self-adjusting and recovering abilities. Regional empirical research on adaptation to global change as advocated by Ge et al. [70] includes research on adaptation to extreme events in the context of global change. Wang et al. [64] evaluated the adaptation of agriculture in response to global warming and drying scenarios in North China. This evaluation is focused primarily on the analysis of climate change with relatively little analysis of the socio-economic aspects. Yasuhara et al. [123] upgraded the methodology to estimate the effects on geo-disasters from combined events, e.g., global warming with increased typhoon and rainfall severity or the occurrence of large earthquakes. Olazabal et al. [124] tracked the progress of governments by analyzing the policies that provide insight into the goals and means of achieving adaptation targets. They identified 226 adaptation policies: 88 at the national level, 57 at the regional/state level, and 81 at the city/metropolitan level.

### 5.2. Adaptation Research in the Field of Natural Disaster Risk

In 1945, White proposed a “series of adjustments” in response to the intensification of flood disasters in the United States. For the first time, attention was given to expand disaster prevention and mitigation from hazards to human behavioral responses to disasters and noted that human behavior can be adjusted to reduce the impact and loss from disasters. This provided new ideas for subsequent comprehensive disaster reduction strategies [125] and established the current natural disaster field to explore the relationships between humankind and the environment by focusing on the impact of extreme events and human responses [17]. Scholars have done more research on the following three aspects of adaptability: disaster risk perception [50], farm-scale uncertainty risk management strategies [85,88], and individual decision making in agricultural systems [126,127,128,129].

To date, there are several evaluations of adaptation measures in the literature, but there are not many cases that evaluate adaptation by constructing an adaptability index system from the perspective of disaster systems. Dhyani and Thummarukuddy [130] proposed a method designed to assess the potential contributions of various adaptation options to improve a system’s coping capacities by focusing attention directly on the underlying determinants of adaptive capacity. Then, they applied this method to expert judgments of six different adaptations that could reduce vulnerability in the Netherlands to increased flooding along the Rhine River.

The adaptation of agriculture is multi-scale and multi-perspective. Research on the adaptability of agriculture to climate change uses several research approaches that consider different scales (plants, sites, fields, farms, regions, countries, and even international) [126,131]. Agriculture is a complex system, and changes within the system are driven by the combined influence of economic, environmental, political, and social forces [132]. Studies have shown that decisions made at different levels of agricultural change are interrelated. Thus, adaptation is the result of individual decisions as determined by the internal forces of the farm family (such as the risk of income loss or environmental perception) and external influences on the power of agriculture systems (such as macroeconomic policies and institutional frameworks) [133].

As an adaptation model of agricultural activities, this is the product of multiple individual decisions (government, agricultural product business, and individual producers) [16,126], government policies, institutional arrangements, and macro social and economic conditions, which are continuously recognized in adaptation research [134]. The actual analysis of the adaptability of the Nile River and Rhine Delta is studied, and two analysis methods of adaptability are provided. One is to obtain the evaluation results through scores of various indicators and the other is to use the synthetic index for the evaluation. The Sahel drought adaptation strategy responds to the following five aspects: precipitation crisis (drought), food supply, livestock management, environmental degradation, and household handling capacity [135]. Therefore, a model of household economic diversity is proposed. This is the single-core structure of the planting industry to the dual-core structure composed of animal husbandry, which then leads to the three-core cycle model of migrant workers and complementary agriculture [94]. Shang [136] analyzed the vulnerability of typical farmers, and Wang et al. [129] analyzed the risk of agricultural drought. They obtained quantitative relationships between income and food production and measures taken to transfer drought risk. This analysis has two prerequisites. One is the factors that affect agricultural drought vulnerability and adaptation, and the other is that these factors can be quantitatively expressed. The factors that meet these two conditions should include the area of conversion of farmland to forests and water conservancy facilities. Wang et al. [121] proposed the vulnerability–adaptation (RA) model to diagnose the adaptation of rain-fed agriculture. The indicators used to evaluate the adaptability are the per capita arable land area, arable land flatness index, irrigation convenience index, irrigation water volume index, per capita food production, per capita proportion of large livestock, and non-agricultural income. Reid et al. [137] interviewed farmers in Boss, Ontario, Canada and focused on four types of farmers and recorded their responses to climate and weather conditions and risks. A broad advance and response management strategy was developed to manage climate risk. Slegers [118] conducted research in Tanzania through questionnaire surveys and interviews and found that farmers are aware of differences in soil types, soil location, land status, and land management practices in drought-vulnerable regions. In fact, farmers have accumulated significant experience in environmental adaptation, which is crucial to the adaptability of the entire agricultural system.

## 6. Discussions

Adaptation to disaster risk is a relatively complex concept. Its complexity is not only reflected in human perception, identification of risks, and the corresponding adaptive management/organizational behaviors but also in the process of adaptation [3,80,97]. The root of this complexity lies in the following. First, the variety of risk factors increases the risk complexity. Various hazard factors and chaining lead to a variety of exposure types under different risk factors, which are fundamentally strengthened. Second, the same hazard factor and its combination hit the same exposure in different regions, which results in different risk levels. The adaptation of the exposure under different risk levels can vary, which further aggravates the complexity. Third, at different spatial scales, the recovery process of the same exposure type will have various focuses that increase its complexity. Fourth, at different stages of historical development, there will be several qualitative changes in the exposures, which lead to a further increase in complexity. Fifth, human perception and the recognition of risks and adaptation differ, causing the process or speed of adaptation to the same disasters of the same intensity to vary, which also greatly increase the complexity.

The complexity of disaster adaptation leads to several influencing factors. The judgment of adaptation should be expressed with multiple indicators. A single indicator inevitably leads to misunderstandings and one-sided conclusions about disaster adaptation. At present, there are several research cases on disaster adaptation, and multiple indicators have been designed from different angles for comprehensive expressions [138,139,140,141]. The combination, division, and relationship determination of disaster adaptation factors gradually reveal the formation of disaster adaptation. At the same time, distinguishing the contribution rates of each influencing factor to disaster adaptation can provide the most powerful basis to strengthen disaster adaptability. To date, the determination of influencing factors for disaster adaptation is based more on the selection of the factors and indicators after understanding the process of disaster adaptation, which has greater subjectivity and uncertainty. Selecting the appropriate method and deciding how to apply reasonable methods to determine the influencing factors of adaptation and the associated contribution rate is the focus of future research.

The 2018 IPCC special report Global Warming of 1.5 °C projects that the climate system is heading off track into the territory of 2.9 to 3.4 °C warming [142]. If this happens, it would take future hydrometeorological hazard extremes well outside the known range of current experience and alter the loss and damage equations and fragility curves of almost all known human and natural systems, placing them at unknown levels of risk [142]. To reinforce SDG 13.1, It also means that it is no longer sufficient to address adaptation in isolation from development planning, and that sustainable socio-economic development, by definition, must include the mitigation of global warming [12].

Looking to the future, it is necessary to track adaptation to identify who is adapting to what, when, where, and why to understand the efficiency of assigned resources and adjust adaptation planning given that information. Adaptation and adaptive capacity should focus on four core issues (Figure 4). The evaluation criteria and adaptation evaluation scales are also shown in Figure 4.

“Adaptation to what” refers to climate change or variation, where adaptation can be a response to adverse effects or vulnerability, which can be a response to the current actual climate or climatic conditions for future predictions. “Who or what adapts” can be an individual, socio-economic department, managed or non-managed system, natural or ecological system, systematic practice, or operation and structure, where each system is distinguished by its adaptability and vulnerability. “How does adaptation occur” can be adaptable to the process or can result in consequences, which can be spontaneous or planned. The above three parts constitute “what is adaptation”, which is a general concept and question that should be answered. The “how good is the adaptation” is an evaluation based on costs, interest, fairness, efficiency, emergency, executability, etc. As a response to countermeasures, the evaluation of choices and measures should be given.

## 7. Conclusions

The adaptation research in the field of natural disasters has focused primarily on socio-economic responses, especially for farm-scale adaptation decision making and risk perception. These socio-economic responses to climate change have accumulated a significant literature basis for further research. However, there are relatively few studies on adaptation evaluation from the perspective of natural disaster systems. The evaluation cases of adaptation for disasters exist primarily as part of vulnerability evaluation. The indicators used at different scales can vary. As adaptability is a local feature, if the scale is too small, the data will be more restricted. If the scale is too large, the sensitivity of the adaptability index used to compare basic units needs to be considered. Learning from current adaptation practices and strengthening them through adaptive governance, lifestyle and behavioral change, and innovative financing mechanisms can help their mainstreaming within sustainable development practices.

Current thinking on disaster adaptation is the result of constructing theories and methods while refining and improving the disaster paradigm over the past few decades [143,144]. The introduction of adaptation in disasters has not been unanimously recognized and is full of controversy. Adaptation requires the integration of multiple disciplines and concepts. Therefore, to summarize the current discussions on disaster adaptation, distinguishing disaster vulnerability, resilience, and adaptation; constructing disaster adaptation assessment models; and exploring regional models for disaster adaptation assessments are important to construct reasonable disaster risk reduction frameworks.

## Figures and Tables

**Figure 1 ijerph-18-11187-f001:**
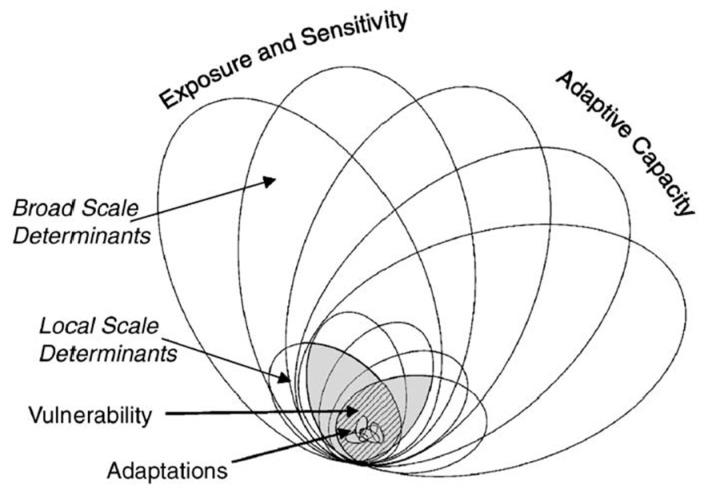
Nested hierarchy model of adaptation, adaptive capacity, and vulnerability.

**Figure 2 ijerph-18-11187-f002:**
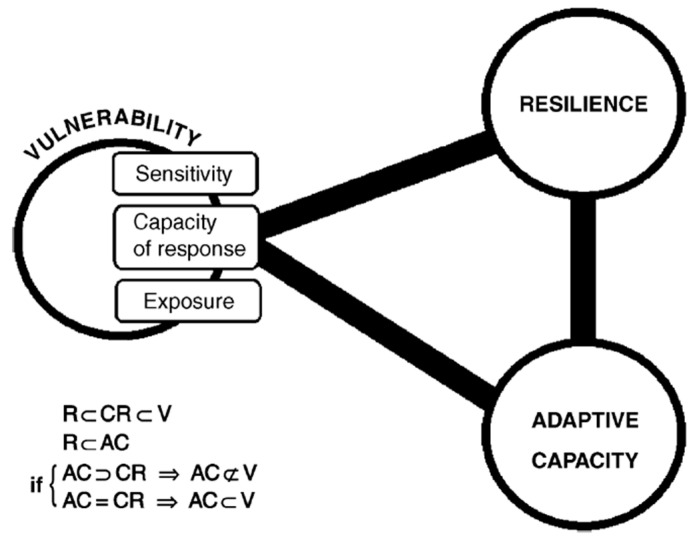
A diagrammatic summary of the conceptual relations between vulnerability, resilience, and adaptive capacity. (The R, V, AC, and CR stand for resilience, vulnerability, adaptive capacity, and capacity of response, respectively.)

**Figure 3 ijerph-18-11187-f003:**
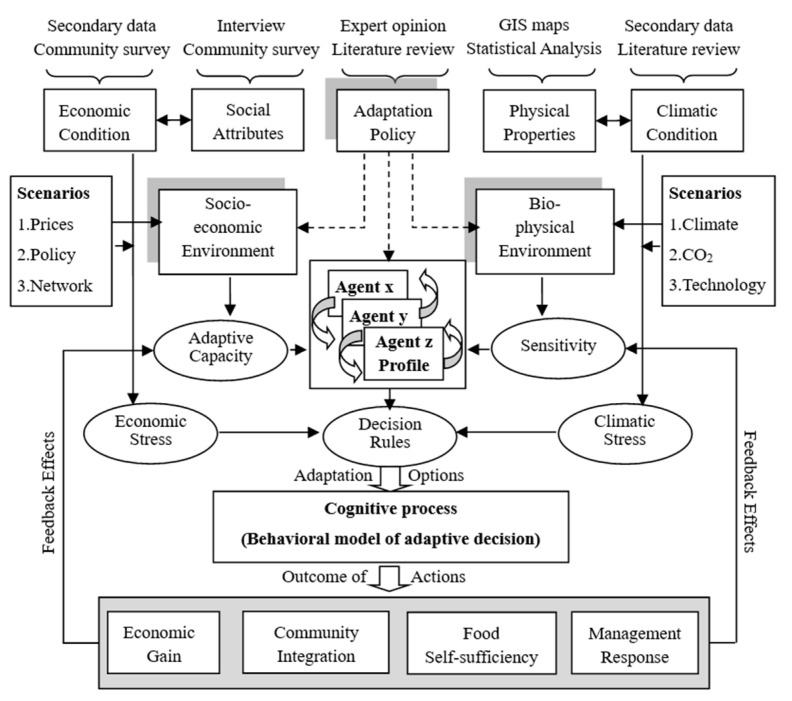
An agent-based inter-vulnerability framework to assess vulnerabilities and adaptations.

**Figure 4 ijerph-18-11187-f004:**
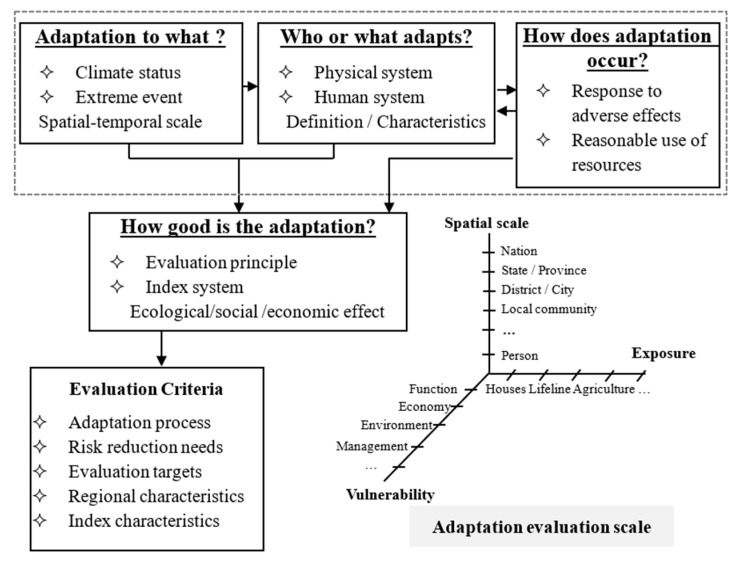
Adaptability constitutes and evaluation framework.

**Table 1 ijerph-18-11187-t001:** Definition of adaptation.

Reference	Definition
(1) Adaptation is a handling capability.	
[16]	Handle the ability of short-term and long-term “possibilities”
[17]	The behavior and features of “system adjustment” can enhance the ability to process external pressure
(2) Adaptation is a response.	
[18]	Ecology–social–economic system: a response to actual and expected climate oscillation and its impact
[19]	Adaptation of climate change refers to a response to human or natural systems on existing or future climate stimuli or influence
[20]	A region or department’s adaptability to climate change relies on many non-climate factors, such as its availability, social and economic policies, cultural and political considerations, individual and public property (economic development and investment levels), and markets or insurance; the adaptability analysis is an important part of the policy response of climate change
(3) Adaptation is a change (adjustment) process.	
[21]	Adaptation includes changes in processes, measures, or structures to reduce or offset potential hazards associated with climate change or to take advantage of the opportunities brought about by climate change, which include reducing the vulnerability of society, regions, or activities to climate change and variability adjustments
[22]	Climate adaptation is a process by which people reduce the negative impact of climate on health and welfare and take advantage of opportunities provided by climate and environmental changes
[23]	Any adjustment measures, whether passive or active, are aimed at reducing the expected adverse effects of climate change
[24]	Climate adaptation countermeasures are adjustment measures taken by individuals for short-term and long-term climate change and extreme weather disasters to enhance the viability of social and economic activities and reduce vulnerability
[25]	Climate change adaptation is defined as the degree to which the implementation, operation process, or structure of the system can be adjusted under possible or actual climate change conditions in the future or the system’s adaptive capacity; adaptation behavior can be spontaneous or planned and can be put into practice in actual processes to handle climate change that has occurred or is expected to occur
[26]	Climate change adaptation includes all human actions or economic structural adjustment measures taken to reduce the vulnerability of all society
[27]	The adjustment of individual organizations and institutional behaviors to reduce the vulnerability of society to the climate change
[18]	The adjustment of the ecological–social–economic system responses to actual or predicted climate change
[28]	Adaptation is a policy option to reduce the negative impact of climate change
[29]	An adjustment of the socio-economic system response to actual or expected climate change
[30]	An adjustment to reduce the risks associated with climate change and vulnerability under its influence to a predetermined level without affecting the existing economic, social, and environmental sustainability
[31]	Adaptation includes both moderating harm and exploiting beneficial opportunities
[32]	Adaptation refers to the process of adjustment to actual or expected climate change and its effects to moderately harm or exploit beneficial opportunities
[33]	Adaptability is a manifestation of adaptation, which is the ability to absorb hazard impacts and to prepare for and recover from them; adaptation in most cases is a proactive action to the anticipated hazards so that potential negative effects or risks can be alleviated in advance
[11]	Incorporate disaster risk reduction measures into multilateral and bilateral development assistance programs within and across all sectors as appropriate, which is related to poverty reduction, sustainable development, natural resource management, environment, urban development, and adaptation to climate change
[34]	Adaptation is a process with varied and changing goals and risk context
[35]	The goal of adaptation is to reduce vulnerability and increase resilience

**Table 2 ijerph-18-11187-t002:** An overview of analysis models and methods of adaptation.

Models and Methods	Characteristics
Scenario-Driven by Climate Change	Represented by the IPCC technical guidelines to evaluate climate change impacts and adaptation countermeasures.
Adaptation Decision Matrix	Suited for analyzing the cost–benefit of adaptation measures.
TEAM	A decision support system software. Suited for assessing the impact and adaptability of climate change in water resources, coastal areas, and agricultural sectors.
Multi-Criteria Evaluation Method	Various adaptation strategies can be compared and evaluated in an orderly and systematic manner.
Agent-Based Modeling Method	A useful policy tool to simulate the effects of different adaptation options toward reducing vulnerability.

**Table 3 ijerph-18-11187-t003:** An overview of five steps of the TEAM model.

Number	Steps
Step 1	Determine the geographic location of the study area.
Step 2	Select possible adaptation strategies and measures.
Step 3	Evaluate the adaptation strategies.
Step 4	Give scores to each standard or indicator.
Step 5	Display the evaluation results.

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
