# Peer review of "Adaptation to Disaster Risk—An Overview"

_ijerph, 2021, doi:10.3390/ijerph182111187_

Round 1
Reviewer 1 Report
This paper provides a comprehensive review of different adaptation models and methods. The literature review is up to date and covers state of the art methods. Unfortunately, the structure of the paper makes it hard to read . There are a few places where tables and figures can be added to clarify the text. The manuscript can also benefit from including a section that describes the impacts of natural disasters. Please refer to the attached pdf version for more specific comments and suggestions.

Author Response
This paper provides a comprehensive review of different adaptation models and methods. The literature review is up to date and covers state of the art methods. Unfortunately, the structure of the paper makes it hard to read. There are a few places where tables and figures can be added to clarify the text. The manuscript can also benefit from including a section that describes the impacts of natural disasters.
Reply:
Thank you for your good comments and suggestions. We are trying hard to improve our manuscript. We appreciate for your warm review work earnestly. The comments are all valuable and very helpful for revising and improving our paper, as well as the important guiding significance to our researches. We have studied comments carefully and have made correction which we hope meet with approval.
Please refer to the attached pdf version for more specific comments and suggestions.
Lines 21-24/ Page 1: The conclusion seems out of context. Please delete or rewrite.
Reply:
Thank you for pointing this out. Special thanks to you for your good comments. We have made modifications according to the Reviewer’s comments (Page 1, line 21-31).
Lines 63/ Page 2: Please add a paragraph to explain the impacts of climate change on natural disasters -e.g. increase in wildfires, flooding, heat waves, etc.
Reply:
We agree with this comment. Thank you for pointing this out. We have added a paragraph to explain the impacts of climate change on natural disasters in the revised manuscript (Page 2, line 72-84).
Lines 69/ Page 2: Maybe just summarize 2.1 and 2.2 and focus on Climate change.
Reply:
Special thanks to you for your good comments. We have summarized 2.1 and 2.2 and made some modifications according to the Reviewer’s comments in the revised manuscript (Page 8-9, line 229-311).
Lines 136/ Page 3: In my personal opinion, this table is not needed. Maybe delete and add any extra information to the text.
Reply:
Thank you for pointing this out. This table is a summary of a large number of references, and it is also organized in time series. After discussion with coauthors, it is clearer to use a table to explain this part of the content. For readers, maybe extra information to the text is not easy to find and sort out. Therefore, table 1 was kept in the revised manuscript (Page 3, line 109-110).
Lines 251/ Page 8: I would suggest adding a table/figure with all the different models/methods.
Reply:
We agree with this comment. Thank you for pointing this out. According to the Reviewer’s comments, a table with all the different models/methods was added in the revised manuscript (Table 2; Page 13, line 450-451; Page 14, line 452).
Lines 351/ Page 9: This is a very good point maybe expand by adding that there are is much uncertainty in applying climate change scenarios.
Reply:
We agree with this comment. According to the Reviewer’s suggestions, some sentences about there is much uncertainty in applying climate change scenarios was added in the revised manuscript (Page 16, line 556-560). We greatly appreciate the reviewers' comment.
Lines 378/ Page 10: Maybe add a chart/table to outline the different steps.
Reply:
Thank you for pointing this out. We agree with this comment. A table to outline the different steps was added in the revised manuscript (Table 3; Page 17, line 617).
Lines 431/ Page 11: It would be nice to list some of these adaptation measures.
Reply:
We agree with this comment. According to the Reviewer’s suggestions, some of these adaptation measures were added in the revised manuscript (Page 17, line 639; Page 18, line 640-642).
Lines 606-610/ Page 15: Please could you find a reference for this?
Reply:
Thank you for pointing this out. According to the Reviewer’s suggestions, a reference for this was added in the revised manuscript (Page 16, line 556-560). We greatly appreciate the reviewers' comment.

Reviewer 2 Report
The paper is really good, has an interesting topic and is well researched.
My minor remarks are related to these points:
- Section 3.2.3. on 'Adaptation and Mitigation' makes strong normative statements:
- In terms of understanding, maybe, could you explain a bit more the sentences in 246/7 'The current focus on adaptation' and in 248/9 '... climate change policies are focused on mitigation...' to avoid the perception of a disagreement between these two statements.
- With regard to content, 249 'the most important thing is adaptation': this strong statement is presented without giving reasons. Maybe you are willing to present the reasons giving by [83] or even to present some reasons given by a more authoritative source for this normative judgement. In general, it might be the case that there are also good reasons for mitigation.
- Lines 265/6: I would like to add 'distribution of costs' to the list. In tandem, distribution of benefits and costs hint to fairness.
- Then, related to Evaluation Standards: 6) in line 312 states that a holistic view of adaptation 'should' be taken. This normative 'should' might be explained a bit more. And, you relate the term 'holistic view of adaptation' to the consideration of other important but non-climate change related aspects. This is not in line with the use of the term 'holistic' by Caney Simon (2012) Just Emissions. Philosophy & Public Affairs 40(4): 255-300. He uses the atomist vs. holistic distinction between using just one principle for adaptation, mitigation, compensation, etc. or using separate principles. And, he uses the isolation vs. integration distinction for considering aspects of climate change in isolation or simultaneously taking other non-climate change related but important (and justice relevant) aspects into account. Therefore, number 6) could benefit from an explanation of the term 'holistic view of adaptation' and providing more reasons for this holistic view.
Author Response
The paper is really good, has an interesting topic and is well researched.
Reply:
Special thanks to you for your good comments. The comments are all valuable and very helpful for revising and improving our paper, as well as the important guiding significance to our researches. We have studied comments carefully and have made correction which we hope meet with approval. Once again, thank you very much for your comments and suggestions.
My minor remarks are related to these points:
Section 3.2.3. on 'Adaptation and Mitigation' makes strong normative statements: In terms of understanding, maybe, could you explain a bit more the sentences in 246/7 'The current focus on adaptation' and in 248/9 '... climate change policies are focused on mitigation...' to avoid the perception of a disagreement between these two statements.
Reply:
Thank you for pointing this out. We agree with this comment. These two expressions may indeed be easy to misunderstand readers, and there may be conflicts. We made some adjustments accordingly in the revised manuscript (Page 7, line 215-228).
With regard to content, 249 'the most important thing is adaptation': this strong statement is presented without giving reasons. Maybe you are willing to present the reasons giving by [83] or even to present some reasons given by a more authoritative source for this normative judgement. In general, it might be the case that there are also good reasons for mitigation.
Reply:
We agree with this comment. Indeed, this strong statement 'the most important thing is adaptation' is inappropriate. The corresponding changes were made in the revised manuscript (Page 7, line 222-227).
Lines 265/6: I would like to add 'distribution of costs' to the list. In tandem, distribution of benefits and costs hint to fairness.
Reply:
Special thanks to you for your good comments. It is really true that distribution of benefits and costs hint to fairness. We have added 'distribution of costs' to the list in the revised manuscript (Page 14, line 462).
Then, related to Evaluation Standards: 6) in line 312 states that a holistic view of adaptation 'should' be taken. This normative 'should' might be explained a bit more. And, you relate the term 'holistic view of adaptation' to the consideration of other important but non-climate change related aspects. This is not in line with the use of the term 'holistic' by Caney Simon (2012) Just Emissions. Philosophy & Public Affairs 40(4): 255-300. He uses the atomist vs. holistic distinction between using just one principle for adaptation, mitigation, compensation, etc. or using separate principles. And, he uses the isolation vs. integration distinction for considering aspects of climate change in isolation or simultaneously taking other non-climate change related but important (and justice relevant) aspects into account. Therefore, number 6) could benefit from an explanation of the term 'holistic view of adaptation' and providing more reasons for this holistic view.
Reply:
We agree with this comment. Thank you for detailed analysis and interpretation. We appreciate for Reviewers’ warm review work earnestly. This normative 'should' was deleted. The expanding is added as follows: Adaptation assessment has become more inclusive over time, linking future climate change with current climate risks and other policy concerns. Adaptation includes well established practices from disaster risk management (e.g., early-warning systems), resource management (e.g., water rights allocation), spatial planning (e.g., flood zone protection), urban planning (e.g., building codes), public health (e.g., disease surveillance), and agricultural outreach (e.g., seasonal forecasts) [92] (Page 15, line 508-514). The term 'holistic' was replaced by ‘systematic’ in the revised manuscript (Page 15, line 508).

Reviewer 3 Report
Dear Authors,
The manuscripts summarized adaptation to disaster risk. It is informative and seems interesting for readers of the journal. However, I would like to ask authors several inquiries.
- Authors use "disasters" but in the test, it seems to be treated as climate change or disasters induced by climate change. I think that disasters are not limited to those related to climate change. Do authors mention about disaster related to climate change only in this manuscript? If no, some corrections is helpful to prevent misunderstanding.
- In section 3, definition of adaptation is described. However, I feel it suitable that this section comes before section 2 because in section 2, development stage of adaptation is described. My suggestion is to switch current section 2 and 3.
- The title of Section 6 "Conclusions and Discussion" is somewhat strange for me. Usually Discussion comes first and then Conclusion comes. Is it possible to separate Discussion and Conclusion and change the order?
Author Response
Dear Authors,
The manuscripts summarized adaptation to disaster risk. It is informative and seems interesting for readers of the journal. However, I would like to ask authors several inquiries.
- Authors use "disasters" but in the test, it seems to be treated as climate change or disasters induced by climate change. I think that disasters are not limited to those related to climate change. Do authors mention about disaster related to climate change only in this manuscript? If no, some corrections is helpful to prevent misunderstanding.
Reply:
Thank you for pointing this out. Special thanks to you for your good comments. In this manuscript, we may focus on disasters and extreme events related to climate change. We have added a paragraph to explain the impacts of climate change on natural disasters -e.g. increase in wildfires, flooding, heat waves, etc according to the Reviewer’s comments (Page 2, line 72-84).
- In section 3, definition of adaptation is described. However, I feel it suitable that this section comes before section 2 because in section 2, development stage of adaptation is described. My suggestion is to switch current section 2 and 3.
Reply:
We agree with this comment. Thank you for pointing this out. We have switched current section 2 and 3 according to the Reviewer’s suggestions (Page 2-13, line 90-311).
- The title of Section 6 "Conclusions and Discussion" is somewhat strange for me. Usually, Discussion comes first and then Conclusion comes. Is it possible to separate Discussion and Conclusion and change the order?
Reply:
Special thanks to you for your good comments. Indeed, it is really true that Discussion comes first and then Conclusion comes. We have made some modifications in the revised manuscript (Page 21, line 779-816; Page 22, line 843-863).

Round 2
Reviewer 1 Report
The new version of the manuscript is much improved and all my suggestions have been addressed. In my opinion, the manuscript is now a lot easier to follow and I like how the abstract has been rewritten.